# High-Performance and Stable Perovskite Solar Cells Using Carbon Quantum Dots and Upconversion Nanoparticles

**DOI:** 10.3390/ijms232214441

**Published:** 2022-11-21

**Authors:** Masfer Alkahtani, Sultan M. Alenzi, Abdulellah Alsolami, Najla Alsofyani, Anfal Alfahd, Yahya A. Alzahrani, Abdulaziz Aljuwayr, Marwan Abduljawad

**Affiliations:** 1National Center for Renewable Energy, King Abdulaziz City for Science and Technology (KACST), Riyadh 11442, Saudi Arabia; 2National Center for Nanotechnology and Semiconductors, King Abdulaziz City for Science and Technology (KACST), Riyadh 11442, Saudi Arabia; 3National Petrochemical Technology Center (NPTC), Material Science Research Institute (MSRI), King Abdulaziz City for Science and Technology (KACST), Riyadh 11442, Saudi Arabia

**Keywords:** carbon quantum dots, upconversion nanoparticles, lithium, perovskite solar cells

## Abstract

Upconversion nanoparticles (UCNPs) and carbon quantum dots (CQDs) have recently received a lot of attention as promising materials to improve the stability and efficiency of perovskite solar cells (PSCs). This is because they can passivate the surfaces of perovskite-sensitive materials and act as a spectrum converter for sunlight. In this study, we mixed and added both promising nanomaterials to PSC layers at the ideal mixing ratios. When compared to the pristine PSCs, the fabricated PSCs showed improved power conversion efficiency (PCE), from 16.57% to 20.44%, a higher photocurrent, and a superior fill factor (FF), which increased from 70% to 75%. Furthermore, the incorporation of CQDs into the manufactured PSCs shielded the perovskite layer from water contact, producing a device that was more stable than the original.

## 1. Introduction

Most nations, especially those whose economies solely rely on natural gas and oil, have begun to seriously worry about conventional energy resources such as oil and natural gas. Conflicts on a local or global scale that result in significant price changes are a source of this concern. Renewable energy is regarded as the ideal substitute for traditional energy sources such as oil and natural gas for this and other reasons [1,2,3]. Solar energy is still the most widely available ecologically acceptable energy source to assure the continued prosperity of the planet. The most common solar cells for converting sunlight into electricity are photovoltaic (PV) cells based on crystalline silicon. These cells offer clean energy for a variety of fascinating applications and have reasonably high operational efficiencies, between 20% and 22% [3,4,5]. Si-based PVs are an established, highly tuned technology with little room for efficiency improvement. However, the extraction of pure silicon from sand requires sophisticated industrial process that consumes a lot of energy and pollutes the environment [4,6]. There are also solar cells that are far more effective, such as those made of gallium arsenide (GaAs), but they are more expensive and prone to degradation [7]. Additionally, organic photovoltaics (OPVs) have received a lot of attention recently, although they are still constrained by their inferior stability and strength compared to inorganic solar cells [8,9].

The extraordinary properties of perovskite materials, such as high quantum efficiency, long diffusion path for charge carriers [10,11,12], high absorption of visible light of the solar spectrum [10,13], and straightforward manufacturing processes [10,14], have made PSCs an attractive alternative, with a PCE exceeding 25.2% in the past decade [4,15,16]. The light-harvesting active layer in PSCs is called perovskite and consists of a compound with a perovskite structure that has an ABX3 (hybrid organic–inorganic) composition [17,18]. To push this field forwards, the engineering of light-harvesting materials of perovskite has been carried out to improve the PCE and increase the photovoltaic performance of PSCs [4,19,20,21,22,23]. Lanthanide (rare-earth) ion- doped upconversion materials (UCNPs) are proposed as promising candidates owing to their optical upconversion luminescence (UL), which can be tuned within the absorption band of the perovskite materials under week near-infrared (NIR) excitations below 10 W/cm^2^ [24,25]. However, the optical performance of UCNPs under weak excitation comparable to sunlight intensity remains a big concern as this could limit their applications in solar cells. To clarify this issue, a key experiment by Xu Chen and his co-authors [25] showed that LiYF_4_:Yb^3+^, Er^3+^ UCNPs at a single-crystal level can efficiently produce a strong UC emission upon excitation by weak irradiation of simulated sunlight (AM 1.5 G) with wavelength higher than 800 nm at a power density as low as 0.1 W/cm^2^. Furthermore, in the same study, LiYF_4_:2%Yb^3+^, 1%Er^3+^ crystals were incorporated into a perovskite solar cell and demonstrated an enhanced PCE of (11%), with a 7.9% enhancement compared to the pristine cell under a simulated solar illumination (AM 1.5 G) at a power density of 0.73 W/cm^2^. Inspired by such an important demonstration, UCNPs have been recently incorporated into different layers of PSCs to improve their photovoltaics performance and PCE [10,26,27,28,29]. In addition to upconversion, UNCPs can also down-convert UV light through a process known as quantum cutting, giving more than one electron per photon and hence improved quantum efficiencies for UV illumination [30]. UCNPs also function as a scattering layer, which can lengthen the route of the light [10]. They can also passivate the grain boundaries of the perovskite layer with fewer imperfections [31], improving PSCs solar performance even further. Consequently, PSCs with UCNPs have outperformed undoped PSCs in terms of PCE, through either direct doping or electrodeposition into the mesoporous layer [29,32].

Recently, promising materials such as carbon quantum dots (CQDs) have also demonstrated a great potential to function as a light converter (down-conversion process), converting ultraviolet and blue photons from sunlight to visible photons. The down-converted light can then be absorbed by the perovskite light-harvesting layer to increase the PCE. Additionally, it was illustrated that passivating the perovskite layer’s grain boundaries using CQDs improved the efficiency of perovskite solar cells and offered great environmental stability [33,34,35]. It was discovered that the photon conversion efficiency of PSCs based on CQD-modified perovskite films improved from 17.59% to 19.38% as a result of the CQD inclusion in the appropriate PSC layers. The hydrophobic CQD molecules that prevented water from coming into contact with the perovskite layer may be responsible for this improvement. Additionally, for four months, the CQD-modified perovskite was stored in an environment with no humidity control. The MAPbI_3_ film still maintained its initial black hue, demonstrating long-term stability [33].

In this study, we present a simple and effective method for producing CQD nanoparticles- and lanthanide ion-doped lithium fluoride-based crystals (YLiF_4_) for modified PSCs. The synthesized CQDs and UCNPs were blended at the optimal mixing ratios with the perovskite layer material and mesoporous layer material (TiO_2_), respectively. The enhanced photovoltaic performance of the CQD- and UCNP-modified PSCs indicated that this is an effective method to integrate promising materials (UCNPs and CQDs) into a PSC composition system to increase its efficiency and long-term stability.

## 2. Results and Discussion

The main goal of this study was to enhance the power efficiency and stability of PSCs by manufacturing a PSC device using two promising materials, namely, CQDs and UCNPs. For this purpose, carbon quantum dots (CQDs) with green emission were synthesized following a microwave synthesis previously reported by Qu, Songnan et al. [8] and then introduced into the fabricated PSC device illustrated in Figure 1a. The CQDs were mixed with the perovskite layer material for passivation at a specific mixing ratio, as described and detailed in the Materials and Methods section. Similarly, the lithium fluoride-based UCNPs (YLiF_4_:Yb,Er) were grown using a solvent thermal method reported in [33] and detailed in the Materials and Methods section. The UCNPs were incorporated into the mesoporous layer material (TiO_2_ nanoparticles) at the best blending ratios, which allowed obtaining the highest power efficiency reported in our previous work [29]. The optimum mixing ratio of the UCNPs with the mesoporous layer material (TiO_2_) was found to be 30%.

Figure 1b demonstrates how the CQDs are expected to passivate and protect the grain boundaries of the perovskite layer, enhance the performance efficiency of the PSCs, and provide excellent environmental stability. Furthermore, the CQDs can also serve as a light converter (down-conversion process) to convert ultraviolet and blue photons to a broad green emission that could be reflected and absorbed by the perovskite light-responsive layer to improve the power efficiency of PSCs. In addition, the upconversion (UC) process in the UCNPs is also important to convert the sunlight’s near-infrared light to visible light, as illustrated in Figure 1c. In this UC process, the UCNPs serve as a light converter to harvest most of the unused near-infrared (NIR) solar light as visible light photons to match the broad absorption band of the perovskite materials.

The structural properties of the synthesized UCNPs and QDs were characterized using a transmission electron microscope (TEM). Figure 2a illustrates dispersed and well-crystalline UCNPs with an average size of 15 nm, as shown in the high- and low-magnification TEM images. The synthesized CQDs also showed ultra-small and crystalline nanocrystals with an average size well below 10 nm, as illustrated in Figure 2b. The surface morphology of the perovskite materials mixed with the ideal concentration of CQDs is shown in Figure 2c,d using scanning electron microscopy (SEM). As a result of the addition of nanoparticles, the perovskite layer exhibits nearly complete surface coverage with larger grains and a crystalline substance. The carbonyl groups (ligands) on the CQDs create an intermediate adduct with Pb^2+^ and are responsible for the increased grain size, which protects the perovskite layer from moisture and degradation [35].

As shown in Figure 3a, a custom designed confocal microscope equipped with a continuous-wave (CW) green (532 nm) laser, an NIR laser (980 nm), optical spectrometers, and single-photon counters was used to assess the optical characteristics of the produced CQDs, UCNPs, and perovskite material film. A drop-cast of the synthesized CQDs, UCNPs, and a perovskite film were placed on a piece of quartz, mounted on the optical apparatus, and exposed to laser radiation. The optical spectrum of the synthesized CQDs under green excitation and captured by the spectrometer revealed a very strong green–red emission, as shown in Figure 3b, which exactly overlapped with the perovskite absorption band. This helped the perovskite to gain more ultraviolet–visible photons from the sun spectrum through the CQDs down-conversion process to improve the PSCs performance as well as their passivation capacities.

The optical upconversion spectrum of the UCNPs under an NIR laser (980 nm) is depicted in Figure 3c. The Er^3+^ ion employed in this study emits visible light in the form of two sharp green and red emission bands of the collected optical spectrum. In the composition designed for this study, the erbium ion (Er^3+^) was chosen because of its distinct visible emission, which exactly matches the perovskite composition absorption band. The physics underlying the upconversion process in UCNPs doped with Er^3+^ is based on the two-photon absorption process. The ytterbium (Yb^3+^) ion, which has a large NIR absorption cross section, functions as a sensitizer for harvesting and transferring the NIR photon energy to the neighboring erbium ion (Er^3+^) via two mechanisms: first, the energy is transferred to the Er^3+^ intermediate and then to its excited states; second, the erbium ion (Er^3+^) undergoes multi-phonon relaxations from its highly excited states to its less excited states, which is followed by radiative emission in the visible spectrum (500–700 nm).

The optical spectrum of the perovskite layer contains the organic and inorganic components needed to be examined as well. Under green excitation, Figure 3d demonstrates that the optical spectrum of the perovskite material reached its maximum at 780 nm. The detected optical emission spectrum was in good agreement with the iodine-containing perovskite composition.

To evaluate the photovoltaic performance of the PSCs with and without the addition of CQDs and UCNPs to the PSC layers, we prepared several PSC devices. The CQD concentration in the perovskite layer was varied with respect to the perovskite material concentrations, as detailed in the Material and Methods section. The mesoporous layer of all devices except the control device was fabricated at a fixed weight ratio of UCNPs/TiO_2_ (30:70 *v*:*v*). To perform the required photovoltaic measurements, all the fabricated devices were given the following names: PSC pristine device (control), PSC device (30% UCNPs), PSC device (30% UCNPs and 100 µL CQDs), PSC device (30% UCNPs and 200 µL CQDs), and PSC device (30% UCNPs and 300 µL CQDs).

To assess the photovoltaic performance of the manufactured PSCs, photocurrent density–voltage curve (J–V) measurements of the devices were carried out experimentally at AM 1.5 G (under one-sun illumination). Table 1 and Figure 4a provide an overview of all the relative photovoltaic properties of the manufactured devices. The PSC device (30% UCNPs and 100 µL CQDs) demonstrated the best photocurrent density (JSC) and PCE, with a 3% increase in the Jsc value from 22.4 to 23.1 (mA/cm^2^) and a 19% enhancement in PCE from 16.57% to 20.44%, as compared to the pristine device and PSC device doped with only 30% UCNPs. The open-circuit voltage (Voc) significantly increased from 1.047 to 1.18 V as the UCNPs were added and the CQD concentration increased. The increase in the photovoltaic performance of the PSC device (30% UCNPs and 100 µL CQDs) can be attributed to the increased incident UV–VIS photons down-converted by the synthesized CQDs and to the NIR photons upconverted by the UCNPs into the perovskite photo-active layer in the fabricated PSCs. Moreover, the down-conversion function of the UCNPs was more efficient and expected to significantly enhance the PSCs efficiency based on the quantum cutting effect. In this process, the UCNPs are expected to give more than one electron per photon and hence improved quantum efficiencies for the UV–VIS sunlight spectrum [30]. Additionally, the CQDs are expected to passivate the grain boundaries of the perovskite layer by increasing the efficiency of the PSCs and providing superior environmental stability [33,34,35]. This is in addition to their optical contribution to the perovskite layer’s absorption band. Additionally, Li doping in the host crystal of the UCNPs allows a quicker electron transport in the mesoporous layer of the PSCs by enhancing surface passivation at the mesoporous layer and perovskite interface [29]. The improved short-circuit current density (JSC), improved PCE, and greater V_oc_ were in excellent agreement with earlier research published in [7,32,33,34,35].

The reverse and forward (J–V) curves of the PSC device (30% UCNPs and 100 µL of CQDs) are shown in Figure 4b. They can be explained by invoking charge accumulation or migration at the perovskite/ETL (mixed with UCNPs) interface. It is possible to decrease the nonradiative recombination at the perovskite/ETL interface and thus effectively ameliorate the anomalous hysteresis behavior of PSCs by improving the electrical characteristics and reducing the surface imperfections of ETL.

The quantum efficiency of the PSC device (30% UCNPs and 100 µL of CQDs), which demonstrated the best performance among the other fabricated devices, was measured and compared with that of the pristine device. The measured quantum efficiency is the ratio of the number of carriers collected by the solar cell to the number of incident photons, limited by the band gap of the perovskite, which is about 1.5 eV. As shown in Figure 4c, the PSC device (30% UCNPs and 100 µL of CQDs) demonstrated a higher incident-photon-to-current conversion efficiency (IPCE) spectrum in the 300–800 nm region in comparison to the pristine device. This enhancement in the IPCE curve in the wavelength range of 350–600 nm can be attributed to the light scattering effect of the synthesized CQDs and UCNPs, which are expected to enlarge the light paths, resulting in enhanced UV–VIS light absorption by the perovskite layer of PSCs [36]. Furthermore, the enhanced QE spectrum also implied a better capability of charge carriers’ collection and a lower charge recombination for the PSC device (30% UCNPs and 100 µL of CQDs) than for the pristine device.

The NIR light-harvesting by UCNPs and its expected contribution to the photovoltaic performance of the PSC device (30% UCNPs and 100 µL of CQDs) were investigated under AM 1.5 G standard sunlight with an 800 nm long-pass filter and compared to the control (pristine device). Under only NIR illumination, the PSC device (30% UCNPs and 100 µL of CQDs) a showed relatively higher performance than the pristine device, as shown in Figure 4d. This additional Jsc current under NIR excitation can be attributed to the optical upconversion luminescence of the UCNPs, in which the low-energy of the NIR photons of the incident light was converted to the high-energy of visible light photons by the UCNPs and absorbed by the perovskite light-harvesting layer, resulting in an additional photocurrent.

The fill factor (FF) of the PSC device (30% UCNPs and 100 µL CQDs) also showed a maximum value of 75% compared with that of the pristine device (Figure 5b). This result indicated that the optimal doping of the UCNPs within the mesoporous layer and the 100 µL addition of CQDs into the PSC devices increased the electron extraction from the perovskite film and therefore enhanced the conductivity of the whole electron transport layer, resulting in a relatively good increase in the FF values.

Finally, it was critical to investigate the stability of the CQD-modified perovskite solar cells in an environment without humidity regulation for several days. For this, we evaluated the stability of the perovskite cell device that presented the best performance (PSC device (30% UCNPs and 100 µL CQDs)) in an ambient environment. The PCEs of the fabricated devices, namely, the PSC device (30% UCNPs and 100 µL CQDs), the PSC device (30% UCNPs), and the pristine device, were evaluated against the aging time, as shown in Figure 5c. The passivated devices with CQDs were more stable than the pristine device. The PCE of the PSC device (30% UCNPs and 100 µL CQDs) remained at 90.1% of its initial value, while the PCE of the pristine device decreased to 72% of its initial value. The improvement in the stability of the PSC device passivated with CQDs is attributed to the passivation of the grain boundaries by the CQDs, which reduced the defect densities in the perovskite film and interacted with the uncoordinated lead ions in the grain boundaries, preventing the degradation from moisture [10,37]. Furthermore, the carbonyl groups (ligands) on the CQDs created an intermediate adduct with Pb^2+^ and were responsible for the increased grain size, which protected the perovskite layer from moisture and degradation [35].

## 3. Materials and Methods

### 3.1. Preparation of the Carbon Quantum Dots (CQDs)

CQDs with green emission were synthesized through microwave synthesis according to a previous method reported by Qu, Songnan, et al. [8]. Briefly, 3 g of citric acid and 6 g of urea were dissolved in 20 mL of DI water and stirred for 5 min. Then, the mixture was heated in a microwave at 750 W for 5 min. A dark-brown cluster was produced. This product was dissolved in ethanol, followed by 3 cycles of centrifugation at 8000 rpm to remove large particles. The resulting yellow-brownish carbon dots in the ethanol solution were left to dry under a fume hood overnight. Different concentrations of carbon dots dissolved in chloroform were prepared for further use.

### 3.2. Preparation of LiYF_4_:Yb, Er UCNP Nanocrystals

An amount of 1.0 mmol of lanthanide ions according to the following mixing ratio for LnCl_3_ [Ln = Y (80.0 wt%), Yb (18.0 wt%), and Er (2.0 wt%)] was placed into a 100 mL three-neck flask containing 10.5 mL of 1-octadecene and 10.5 mL of oleic acid. A transparent yellow solution was obtained by progressively heating the combination to 150 °C in a Schlenk line for 50 min under argon flow. A mixture of 5.0 mL of a methanol solution containing 2.5 mmol of LiOH.H_2_O and 10.0 mL of a methanol solution containing 4.0 mmol of NH_4_F was added drop by drop into the solution after it had cooled to 50 °C. After that, the mixture was kept at 50 °C for 40 min while being vigorously stirred. After 20 min, methanol and water were removed from the solution by progressively raising the temperature to 150 °C. The reaction mixture was subsequently heated to 300 °C for 1.5 h while argon flowed through it. The mixture was cooled to room temperature once the reaction was finished, and the synthesized YLF UCNPs were then collected, washed three to four times with ethanol, and then redispersed in 10 mL of chloroform.

### 3.3. Producing Ligand-Free UCNPs for the Solar Cell Experiment

To combine UCNPs with TiO_2_ nanoparticles in the mesoporous layer, ligand-free and hydrophilic UCNPs were prepared. For this, 1.0 mL of the LiYF_4_:Yb,Er UCNPs with oleate caps was dissolved in 40 mL of ethanol solution at a pH of 1. Concentrated hydrochloric acid was used to modify the pH. To strip off the oleate ligands, the solution was sonicated for an hour. Following the completion of sonication, the nanoparticles were collected by centrifugation at 14,500 rpm for 30 min. After that, the particles were washed in a solution of ethanol/water (1:1 *v*/*v*) for three times. The Ln-UCNPs were re-dispersed in 100% ethanol after becoming oleate-free [29].

### 3.4. Preparation of the Mesoporous Layer Solution

A commercial TiO_2_ paste (Dyesol 30NRT, Dyesol) was used with the synthesized LiYF_4_:Yb,Er UCNPs to create the mesoporous layer of the manufactured PSCs. The optimal mixing ratio described in our earlier work [29] (UCNPs:TiO_2_ = v(UNCPs) × 100/v(TiO_2_), *v*/*v*, x = 30) was achieved by dissolving 1 mmol of ligand-free Ln-UCNPs in 1.0 mL of pure ethanol before mixing it with the TiO_2_ paste. Except for the pristine sample, each sample’s mesoporous layer contained 30% UCNPs. To maintain a concentration of 0.1 g/mL, the solutions were ultrasonically diluted in various concentrations of 100% ethanol. Prior to the experiments, the solution was maintained under stirring overnight after the sonication was finished.

### 3.5. Perovskite Solar Cell Device Fabrication

Sample substrates made of fluorine-doped tin oxide (FTO) glasses were cleaned by sonication in soap for 25 min, DI water for 10 min, ethanol for 10 min, and acetone for 10 min. The FTO substrates were cleaned and then subjected to a 30 min UV–ozone treatment.

Compact layer preparation: A thin compact TiO_2_ layer was deposited on top of all FTO substrates by spin coating. The solution used in this coating process contained 0.5 mL of titanium isopropoxide, 0.5 mL of acetylacetonate, and 9.0 mL of ethanol. The solution was dropped on the substrates and spin-coated for 30 s at 2500 rpm, followed by 30 min of annealing at 460 °C. After completing this procedure, the mesoporous layer mixed with 30% of the UCNPs was added on top of the compact layer by spin coating the TiO_2_/UCNP mixture at 5000 rpm for 25 s, followed by 30 min of annealing at 500 °C.

A mixture of the following substances was used to prepare a perovskite precursor solution. For this, 18 mg of methylammonium bromide, 720 mg of lead(II) iodide, 247 mg of formamidine bromide, 21 mg of cesium iodide, 60 mg of lead(II) bromide, 240 μL of dimethyl sulfoxide (DMSO), and 960 μL of dimethylformamide (DMF) were combined and then gradually heated to 80 °C for 15 min to form a homogeneous triple-cation composition. For the precursor of the CQD-modified perovskite film, different amounts of CQD solutions (100, 200, and 300 μL) were added to obtain mixtures with different concentrations of CQDs. Afterwards, 50 µL of the perovskite precursor solution was spin-coated at two speeds, i.e., at 1100 rpm for 15 s and then at 5000 rpm for 25 s for each device, namely, the PSC device (30% UCNPs and 100 µL CQDs), the PSC device (30% UCNPs and 200 µL CQDs), and the PSC device (30% UCNPs and 300 µL CQDs). After applying 200 µL of chlorobenzene to the substrates for 15 s to eliminate any remaining DMSO and DMF from the precursor films, the substrates were annealed at 120 C for 40 min on a hotplate to create well-crystalline triple-cation perovskite layers.

After that, a hole transfer layer (HT), spiro-MeOTAD, was deposited on top of the triple-cation perovskite layers by spin coating for 25 s at 5000 rpm. Following that, a unique shadow mask was used to thermally deposit an 80 nm-thick gold layer on top of the spiro-MeOTAD layers in a high vacuum. Finally, the manufactured PSC devices with an active area of 0.1 cm^2^ (0.25 × 0.4 cm^2^) were ready for the photovoltaic performance tests.

## 4. Conclusions

We synthesized and introduced high-quality CQDs and UCNPs into the perovskite and mesoporous layers of PSCs. The manufactured PSCs showed an enhancement in power conversion efficiency (PCE) from 16.57% to 20.44%, a higher photocurrent, and a superior fill factor (FF), increasing from 70% to 75%, in comparison to the pristine PSCs at the ideal mixing ratios of the synthesized CQDs and UCNPs. Furthermore, the PSCs doped with CQDs and UCNPs showed better stability than the pristine device. The reported results suggest that CQD added to PSCs form a protective layer that prevents the contact of the perovskite film with water (moisture), thereby enhancing the stability of the PSCs for efficient renewable energy applications.

## Figures and Tables

**Figure 1 ijms-23-14441-f001:**
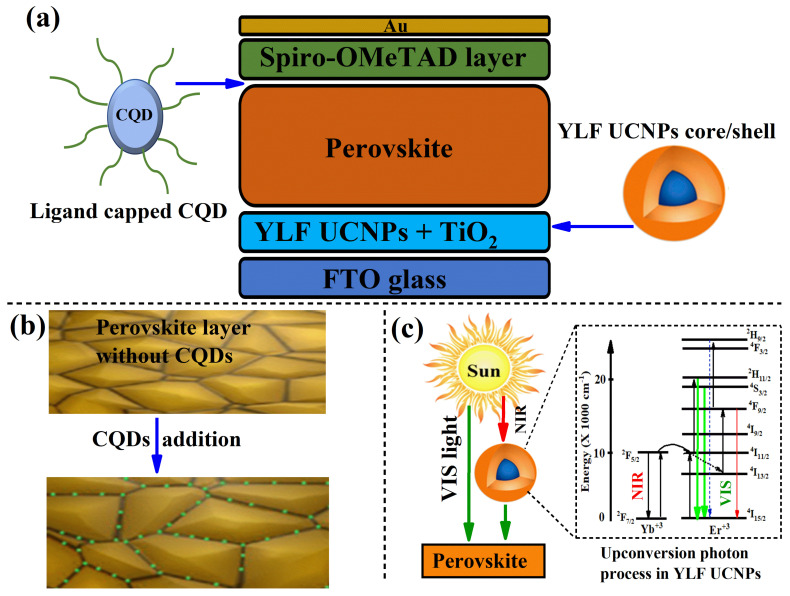
(**a**) Schematic diagram of the PSC layers where the ligand-capped CQDs are mixed at different concentrations with the perovskite layer, while the UCNPs are embedded in the mesoporous layer. (**b**) Illustration diagram of the proposed passivation mechanism of CQDs in the PSC devices. (**c**) Schematic diagram of the upconversion process of converting wasted NIR photons of sunlight to visible light by the UCNPs.

**Figure 2 ijms-23-14441-f002:**
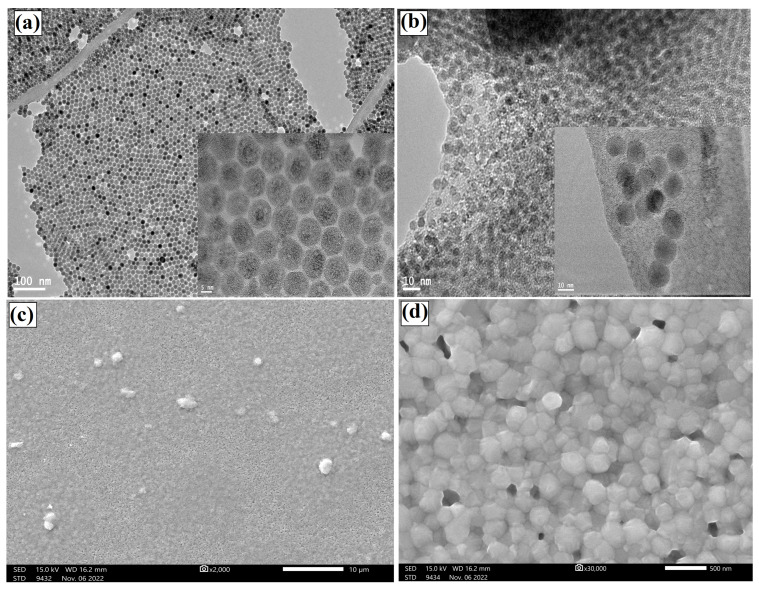
Structural characterization of the synthesized UCNPs and CQDs. (**a**) High- and low-magnification TEM images of the synthesized UCNPs. (**b**) TEM images of the synthesized CQDs with an average size below 10 nm. (**c**) Low-magnification SEM image of the CQDs mixed with the perovskite materials and spin-coated on a quartz slide. (**d**) High-resolution SEM image of the prepared hybrid materials.

**Figure 3 ijms-23-14441-f003:**
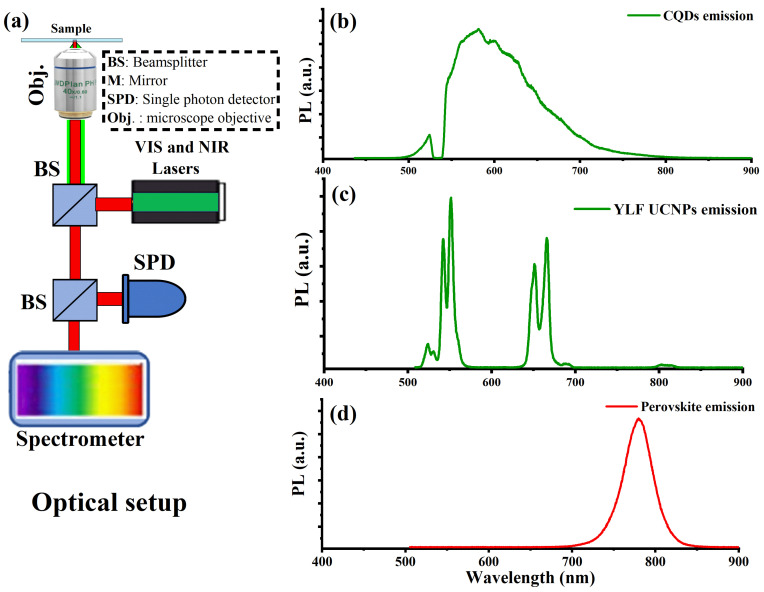
(**a**) Schematic of a home-made confocal microscope developed and fitted with near-infrared and visible lasers for photoluminescence (PL) measurements. The CQD spectrum, measured directly from the CQD nanoparticles, is shown in the optical spectrum in (**b**). (**c**) Visible emission of UCNPs, which consists of two emission bands in the red and green regions that correspond to the Er^+3^ optical transition under 980 nm laser stimulation. Additionally, (**d**) displays the PL spectra of the perovskite layer, centered at 780 nm under green excitation.

**Figure 4 ijms-23-14441-f004:**
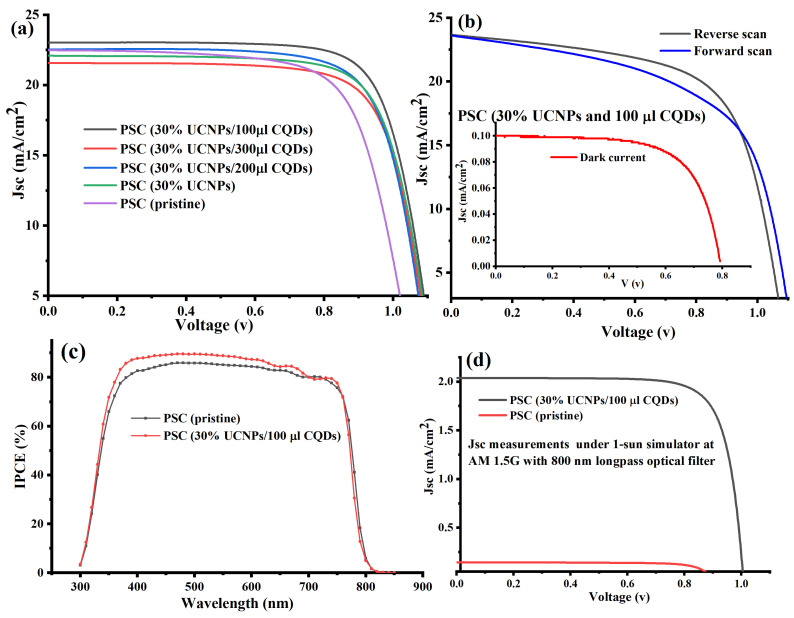
(**a**) J–V characteristic curves for PSCs with and without UCNP and CQD levels integrated inside the mesoporous and perovskite layers, as measured under AM 1.5 G. (**b**) Reverse and forward scans of the (J–V) curves of the best fabricated device (PSC device (30% UCNPs and 100 µL CQDs)). ((**b**) inset) Dark-current measure for the PSC device (30% UCNPs and 100 µL CQDs). (**c**) Quantum efficiency (IPCE) spectra of the PSC device (30% UCNPs and 100 µL CQDs) and the pristine device. (**d**) J–V characteristics measured under NIR irradiation with an 800 nm long-pass filter for the PSC device (30% UCNPs and 100 µL CQDs) and the pristine device.

**Figure 5 ijms-23-14441-f005:**
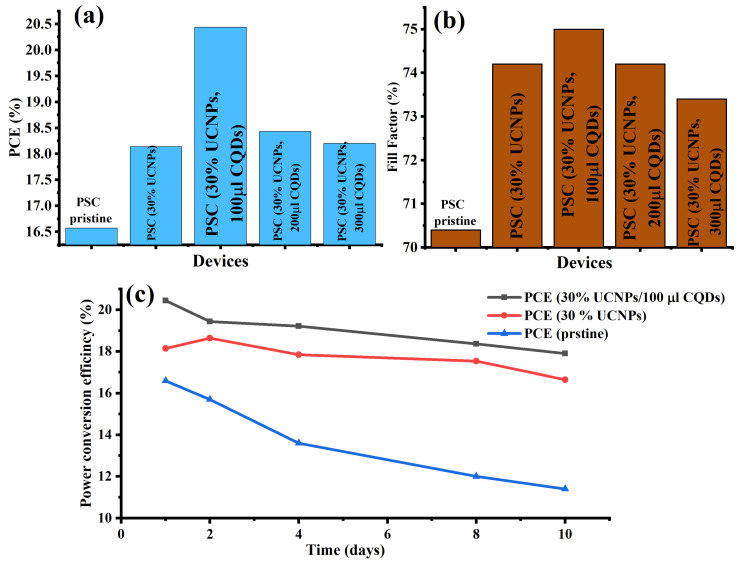
(**a**) PCE values of the synthesized PSCs in relation to the UCNP and CQD quantities integrated inside the mesoporous and perovskite layers. The enhancement of the constructed devices’ fill factor (FF) is shown in (**b**). (**c**) PCE values of the manufactured PSC devices with and without UCNPs and CQDs in ambient air for several days without regulating the humidity.

**Table 1 ijms-23-14441-t001:** A summary of the photovoltaic parameters of the manufactured devices.

Sample	PCE (%)	Voc (V)	Jsc (mA/cm^2^)	FF (%)
PSC device (30% UCNPs and 100 µL CQDs)	20.44	1.18	23.1	75
PSC device (30% UCNPs and 200 µL CQDs)	18.43	1.104	22.55	74.2
PSC device (30% UCNPs and 300 µL CQDs)	18.2	1.101	22.52	73.4
PSC device (30% UCNPs)	18.14	1.107	22.08	74.2
PSC device (pristine)	16.57	1.047	22.47	70.4

## Data Availability

The data presented in this study are available on request from the corresponding author.

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
