# Peer review of "High-Performance and Stable Perovskite Solar Cells Using Carbon Quantum Dots and Upconversion Nanoparticles"

_ijms, 2022, doi:10.3390/ijms232214441_

Round 1

Reviewer 1 Report (Previous Reviewer 1)

1) The authors response that the UCNPs can be excited by a very low NIR excitation (a wavelength of 10 W/cm2). Whereas, a sunlight intensity with total wavelength is 100 W/cm2, considering the small portion of NIR light of 800-1000nm, the NIR intensity of sunlight is much lower than that of 10 W/cm2, so I still can not believe that UCNPs can be excited by sunlight. And I have not seen any publications about low laser intensity excite UCNPs, can you show the publications?And the external quantum efficiency of devices can verify the enhancements of light absorption in NIR region and the enhanced Jsc, can you provide the EQE spectra of devices with QDs and without QDs.

2) For SEM of hybrid films with QDs, I can not see the QDs, can you show magnified images, or mark the QDs in image. 

Author Response

Reviewer 2 Report (Previous Reviewer 2)

The manuscript was revised and improved by the authors. It can be accepted in this form.

Author Response

Thank you very much!!

Round 2

Reviewer 1 Report (Previous Reviewer 1)

I do think the authors provide enough experiments and demonstration to greatly improve the manuscript quality compared with that of old versions. 
However, about the demonstration of EQE results, it is still unclear. I strongerly suggest the author refer to the EQE demonstration in "Nanomaterials, 2020, 10(5), 944.", please add the related demonstrations and cite the refrence.

Author Response

This manuscript is a resubmission of an earlier submission. The following is a list of the peer review reports and author responses from that submission.

Round 1

Reviewer 1 Report

The manuscript of “High-performance and stable perovskite solar cells using car-2 bon quantum dots and upconversion nanoparticles” reports a hybrid Perovskite polymer solar cells that doped with upconversiton nanoparticles and carbon quantum dots.

There are some reports about UCNPs doping PSCs to increase the photovoltaic performances, but the enhancements are not evident. One big obstacle is UCNPs should be excited by laser light, whereas, the intensity of sunlight is too low to excite the UCNPs. So I suspect why the PCE enhanced so much under sunlight simulator, in theory, UCNPs can not emit light under sunlight excitation.

Besides, there is not enough evidence to prove the conclusions. Some necessary and basic characterizations are not provides, such as TEM of UCNPs and CQDs, SEM of hybrid films with QDs.

Therefore, I recommend reject this manuscript. I also suggest the authors added necessary characterizations and resubmit this manuscript. 

Reviewer 2 Report

The manuscript reports on improvements in the stability and efficiency of perovskite solar cells through the use of Upconversion nanoparticles and carbon quantum dots. Improved power conversion efficiency, higher photocurrent, and a superior fill factor are reported, as well as improved environmental stability.
Overall, the manuscript is clear, relevant for the field, and presented in a well-structured manner.

Please consider the following points to improve the manuscript:

(1) Errors with spaces, punctuation and numbering disturb reading and must be corrected.

(2) This reference might be added:
Mirsafaei, M.; Luis, A.; Cauduro, F.; Kunstmann-olsen, C.; Michael, A.; Hassing, S.; Hedegaard, M.A.B.; Rubahn, H.; Adam, J.; Madsen, M. Periodically arranged colloidal gold nanoparticles for enhanced light harvesting in organic solar cells. Photonics Sol. Energy Syst. VI 2016, 9898, 989810.

(3) The manuscript would benefit from an outlook at the end that places the significance of the research in a larger context and addresses future steps or research questions.

Reviewer 3 Report

Masfer et al. described the mixing of Upconversion nanoparticles and Carbon quantum dots for efficient and stable PSCs with an overall efficiency of around 20%. It is interesting to see the higher-efficiency devices by combing the two materials mentioned above in an appropriate concentration, but there are a few things that need to be added in the current version of the manuscript before publication.
1) Revision of this manuscript is required in terms of grammatical mistakes. As you can see on line 53, "In in." and TIO2, the I is capital. The authors need a very careful revision of the manuscript. 
2) dark currents should be added to the manuscript.
3) The reverse and forward scan should be in J-V curves. It is hard to see the device's stability using a reverse scan.
4) What is the role of ligands on the surface of CQDs, Authors should explain in detail the possibility of having a larger surface covered by ligands of CQDs.
5) What is the effect of CQDs on the perovskite crystal size. There is no SEM or AFM image as can be seen in figure 1, authors tried to show CQDs over the perovskite layer.
6) Why the efficiency of these devices increased, An extensive explanation is required.
